# An Enzyme-Linked Immunosorbent Assay for the Detection of Mitochondrial DNA–Protein Cross-Links from Mammalian Cells

Wenyan Xu [1] and Linlin Zhao [1,2,*]

1 Department of Chemistry, University of California Riverside, Riverside, CA 92521, USA
2 Environmental Toxicology Graduate Program, University of California Riverside, Riverside, CA 92521, USA
* Correspondence: linlin.zhao@ucr.edu; Tel.: +1-951-827-9081

**Abstract:** DNA–Protein cross-links (DPCs) are cytotoxic DNA lesions with a protein covalently bound to the DNA. Although much has been learned about the formation, repair, and biological consequences of DPCs in the nucleus, little is known regarding mitochondrial DPCs. This is due in part to the lack of robust and specific methods to measure mitochondrial DPCs. Herein, we reported an enzyme-linked immunosorbent assay (ELISA)-based method for detecting mitochondrial DPCs formed between DNA and mitochondrial transcription factor A (TFAM) in cultured human cells. To optimize the purification and detection workflow, we prepared model TFAM-DPCs via Schiff base chemistry using recombinant human TFAM and a DNA substrate containing an abasic (AP) lesion. We optimized the isolation of TFAM-DPCs using commercial silica gel-based columns to achieve a high recovery yield for DPCs. We evaluated the microplate, DNA-coating solution, and HRP substrate for specific and sensitive detection of TFAM-DPCs. Additionally, we optimized the mtDNA isolation procedure to eliminate almost all nuclear DNA contaminants. For proof of concept, we detected the different levels of TFAM-DPCs in mtDNA from HEK293 cells under different biological conditions. The method is based on commercially available materials and can be amended to detect other types of DPCs in mitochondria.

**Keywords:** abasic sites; DNA lesions; DNA damage; protein DNA adducts; DNA repair; mitochondrial DNA (mtDNA); mitochondrial transcription factor A (TFAM)

## 1. Introduction

DNA–protein cross-links (DPCs) are cytotoxic DNA lesions that pose a significant threat to genomic stability. In the nucleus, the formation of DPCs has been documented in a variety of processes, such as DNA damage by reactive chemicals, DNA repair, and topological transformations [1–3]. DPCs can also form during chemotherapy and radiation therapy as means to kill cancer cells [4,5]. Consequently, DPCs represent a diverse class of DNA lesions, which differ in the protein component, the flanking DNA structure, and the cross-linking chemistry. Several recent reviews have summarized the different types of DPCs and their repair pathways in detail [2,3]. In general, the repair of DPCs involves proteolytic digestion of the protein component of DPCs, with additional repair activities varying depending on specific types of DPCs [2,3]. Certain types of DPCs can also be resolved by the enzymatic hydrolysis of the covalent linkage between DNA and protein [3]. A prominent example is topoisomerase-mediated DPCs with their cognate repair enzymes, tyrosyl-DNA phosphodiesterases [2,3]. Despite the significant progress in the understanding of the formation and repair of DPCs with nuclear DNA (nDNA), limited knowledge exists regarding mitochondrial DPCs. Although mitochondrial DNA (mtDNA) accounts for merely 1–5% of the total cellular DNA by weight [6], it is essential for mitochondrial and cellular functions. This is because mtDNA not only encodes 37 essential genes but also acts as a cellular genotoxic stress sentinel [7]. mtDNA instability in the forms of mutations,

deletions, ablation, or depletion has been associated with a broad spectrum of human disorders and aging [8–10].

Reliable and quantitative evaluation of DPCs is key to the understanding of the formation and resolution of DPCs, and to the evaluation of the efficacy of cancer therapeutics that exploit the cytotoxicity of DPCs. A number of methods have been developed for detecting nuclear DPCs. Historically, ultracentrifugation-based methods have been used to isolate topoisomerase DPCs [11,12]. The major limitations of these methods are low throughput and the requirement of large amounts of starting materials (typically 2 to $10 \times 10^6$ cells per sample) and specialized equipment. Another method using a commercial guanidine-detergent lysing solution has been shown to be effective in removing noncovalent proteins, followed by mass spectrometry-based detection [13]. However, the method requires a relatively large amount of starting materials and extensive handling, and is also limited in throughput. Recently, a RADAR (rapid approach to DNA adduct recovery) assay based on immunodetection has been developed and validated for detecting topoisomerase DPCs in human and bacterial cells and gyrase-DNA adducts in *Escherichia coli* [14–16]. The method is relatively simple and requires as low as 60 ng of DNA (approximately $1 \times 10^4$ cells) per well, which is potentially amendable to other types of DPCs and high-throughput workflows. In spite of the improved detection of nuclear DPCs, few methods are available for detecting mitochondrial DPCs. A majority of the purification procedures developed for nuclear DPCs are not suitable, at least not without optimization, owing to the overall low abundance (~1% by weight) of the mtDNA in a cellular DNA sample [6]. The low amounts of mtDNA also require careful validation of the existing detection methods. The best-studied human mitochondrial DPCs are topoisomerase 1 DPCs [17–19], and the other known mitochondrial DPCs form from the tyrosyl-DNA phosphodiesterase 1 (TDP1) H493R variant [18]. The detection of these DPCs has been achieved using a dot-blotting method based on immunodetection using a specific antibody for the mitochondrial isoform of topoisomerase 1 [17,18]. The assay requires microgram DNA input, which could limit applications of the method in a high-throughput format.

We intended to develop a simple and accessible approach without a lengthy sample workup for mitochondrial DPCs. Herein, we reported an enzyme-linked immunosorbent assay (ELISA)-based approach to detect mitochondrial DPCs. The development of the method is motivated by our recent discovery of the role of mitochondrial transcription factor A (TFAM) in DNA strand scission at abasic (AP) sites [20]. The reaction between TFAM and AP sites proceeded via Schiff base chemistry, forming TFAM-DPCs (chemical structures shown in Figure 1). We chose an ELISA-based method due to its high throughput and the relatively low amount of samples required. The latter reason is particularly important for mtDNA-associated DPCs because of the low abundance of mtDNA in cellular samples and the low stoichiometric ratios of cross-linked proteins relative to mtDNA. TFAM-induced DPCs (and perhaps other proteins) are a heterogenous mixture of DPCs mediated by different Lys residues [21]. For this reason, although mass spectrometry-based approaches are powerful in detecting specific types DPCs [22], the heterogeneity of TFAM-DPCs would make mass spectrometry-based quantitation challenging due to different recovery yields and ionization efficiencies of the peptides. TFAM is one of the most abundant proteins in mitochondrial nucleoids and binds to DNA with low nM affinity [20]. To overcome the challenge of a high stoichiometry ratio of TFAM/DNA, we optimized the DPC purification and detection procedures to maximize the specificity for TFAM-DPCs, minimizing the interference from residual noncovalently associated TFAM during the sample workup. Because of the low overall amount of mtDNA relative to the total cellular DNA, we also optimized the mtDNA isolation procedure to eliminate nearly all contaminating nDNA fragments. For proof of principle, we detected an elevated level of TFAM-DPCs in HEK293 cells when AP sites were induced in mtDNA. Our data imply the involvement of TFAM in AP sites processing in mitochondria. The method is amendable to detecting other mitochondrial DPCs and will facilitate the understanding of the formation and repair of mitochondrial DPCs.

## 2. Materials and Methods

### 2.1. Chemicals and Materials

Chemicals were from Fisher Scientific (Pittsburgh, PA, USA) or Research Products International (Mt. Prospect, IL, USA) and were of the highest grade available. Oligodeoxynucleotides were synthesized and HPLC-purified by Integrated DNA Technologies (Coralville, IA, USA). Recombinant human TFAM was prepared as described previously [20]. Uracil-DNA glycosylase (UDG, M0280L) was from New England Biolabs (Ipswich, MA, USA). TFAM primary antibodies were from Novus Biologicals (NBP1-71648, Centennial, CO, USA) and Cell Signalling Technology (7495S, Danvers, MA, USA). Horseradish Peroxidase (HRP)-conjugated secondary antibodies were from Novus Biologicals, LLC (HAF007) and Santa Cruz Biotechnology (sc-2357, Dallas, TX, USA). Fast kinetic HRP substrate, 3,3′,5,5′-tetramethylbenzidine (TMB, ab171524), was from Abchem (Kenosha, WI, USA). The SuperSignal™ ELISA chemiluminescent substrate (37069) was from Thermo Scientific (Waltham, MA, USA). Bio-Spin® 6 column (7326200) was from Bio-Rad (Hercules, CA, USA). The DNA Clean & Concentrator-5 kit (D4014) was from Zymo Research Corp (Irvine, CA, USA). The Monarch™ genomic DNA purification kit (T3010), Monarch™ PCR & DNA cleanup kit (T1030), and λ DNA (N3012) were from New England Biolabs. Turbonuclease (T4330) was from Sigma-Aldrich (St. Louis, MO, USA). HEK293 Tet-On cells (631182) were from Takara Bio Inc. (Kusatsu, Shiga, Japan). ELISA high-binding 96-well plates were from Greiner (cat No. 650061), and 96-well tissue culture plates were from BrandTech (Essex, CT, USA; cat No. 781962 for the TMB substrate and 781965 for the chemiluminescent substrate).

### 2.2. Preparation of Model TFAM-DPCs

A deoxyuridine (dU)-containing single-stranded DNA oligomer (AACCCTAACACCA GCCTAACCAGATTTCAAATTTTATCTTTTGGCGGTATGCACTTTTAACAGTCACCCCCC <u>dU</u>AACTAAC) was annealed with a complementary oligomer (GTTAGTTAGGGGGGT-GACTGTTAAAAGTGCATACCGCCAAAAGATAAAATTTGAAATCTGGTTAGGCTGGT GTTAGGGTT) at 95 °C for 15 min in the presence of 30 mM MES (2-(N-morpholino)ethanesulfonic acid) buffer (pH 6.5), 100 mM NaCl, and 1 mM EDTA, followed by cooling on the heat block overnight. An aliquot (μL) was resolved in a 3% agarose gel to confirm the successful annealing. To convert dU to AP sites, the annealed duplex DNA substrate (3 nmol) was incubated with UDG (6 units) at 37 °C for 6 h in the presence of 20 mM HEPES (4-(2-hydroxyethyl)-1-piperazineethanesulfonic acid) buffer (pH 8.0), 1 mM DTT (dithiothreitol), and 1 mM EDTA (ethylenediaminetetraacetic acid). To confirm the digestion, an aliquot (1 μL) of the UDG digestion mixture was diluted with water and incubated with 0.3 M NaOH at 65 °C for 30 min. The alkaline cleavage products containing single-strand breaks at AP sites were resolved on a 38 × 30 cm 18% Urea (7M)-TBE-PAGE. The gel was stained in 100 mL of 1×TBE buffer containing 1× SYBR Gold for 30 min at 22 °C, followed by destaining with 100 mL of 1×TBE buffer at 22 °C for 30 min. The AP site-containing DNA was purified through phenol/chloroform extraction and a Bio-spin 6 column. TFAM-DPCs were prepared through a reaction containing 1 μM purified AP-DNA, 8 μM TFAM, 20 mM HEPES (pH 7.4), 300 mM NaCl, 10 mM EDTA, and 25 mM $NaBH_3CN$ at 37 °C for 24 hours. $NaBH_3CN$ reduces the Schiff base intermediates formed between TFAM and AP sites in situ and results in stabilized TFAM-DPCs. The product was resolved with a 3% agarose gel containing 0.1% SDS to verify the formation of DPCs.

### 2.3. Purification of TFAM-DPCs from the In Vitro Reaction

For the phenol/chloroform extraction, the reaction mixture was mixed thoroughly with an equal volume of phenol/chloroform/isoamyl alcohol and centrifuged at $16,000\times g$. The aqueous phase was collected, and the organic phase was washed with another 1× volume of HEPES/EDTA (10 mM/1 mM) buffer. The two aqueous fractions were combined, followed by buffer exchange into a HEPES/EDTA solution with a Bio-spin 6 column. Purifications with commercial kits were performed following the manufacturers' recommended procedures with modifications. For the DNA Clean & Concentrator-5 kit, the

reaction mixture was mixed with a 7× volume of the DNA binding solution and loaded onto the column, followed by centrifugation at 13,000× *g* for 30 s. The column was washed twice with the washing buffer. A 10 μL solution (preheated to 60 °C) containing 20 mM HEPES (pH 7.4), 0.1 mM EDTA, and SDS (0.1% *w/v*) was applied to the column. The column was incubated at 22 °C for 1 min, followed by elution of TFAM-DPCs using centrifugation at 13,000× *g* for 1 min. The load–wash–elute cycle was repeated thrice to achieve a >90% yield of TFAM-DPCs. For the purification using the NEB genomic DNA purification kit, the sample was mixed with an equal volume of cell lysis buffer, followed by mixing with 500 μL of the DNA-binding solution from the kit. The sample was loaded onto the column followed by centrifugation steps at 3000× *g* for 3 min, followed by 13,000× *g* for 1 min. The dry column was washed twice using the wash buffer from the kit. A 40 μL solution (preheated to 60 °C) containing 20 mM HEPES (pH 7.4), 0.1 mM EDTA, and SDS (0.1% *w/v*) was applied to the column. The column was incubated at 22 °C for 1 min, followed by elution of TFAM-DPCs by centrifugation at 13,000× *g* for 1 min. Purification with Monarch® PCR & DNA cleanup kit was performed following the manufacturer's protocol.

### 2.4. Quantification of TFAM-DPCs and TFAM by ELISA

Varying amounts (0–8 fmol) of TFAM-DPCs or recombinant TFAM were loaded in wells containing 150 μL of coating solution as specified. The plate was incubated at 22 °C for 16–18 h in the dark. The solution was aspirated, and the wells were washed with a 1× TBST (tris-buffered saline with 0.1% Tween 20) solution five times. Blocking was carried out using 3% (*w/v*) BSA in a 1× TBST solution at 22 °C for 1 h. A 150 μL solution was applied to each well, which contained a TFAM primary antibody (1:500 dilution with 1× TBST solution containing 3% (*w/v*) BSA). The plate was incubated at 4 °C for 22–24 h. After washing with a 1× TBST solution five times, each well was incubated with a 150 μL solution containing an HRP-conjugated secondary antibody (1:1000 dilution) at 22 °C for 1 h. After the incubation, the wells were washed again with a 1× TBST solution five times. An HRP substrate (150 μL) was added to each well. For the TMB fast kinetic substrate, the plate was incubated at 37 °C for 40 min before the readings were taken with a plate reader equipped with a filter for absorbance at 650 nm. For the chemiluminescent substrate, the readings were acquired immediately after adding the substrate under the following parameters: integration of 5 sec, gain value of 250, and a read height of 1 mm. Data were acquired on a Synergy H1 plate reader. The lowest detectable amounts of TFAM-DPCs and TFAM were defined as fmol of TFAM (or TFAM-DPCs) when the signal intensity was three times of that of the background.

### 2.5. Standard Curve and Recovery of TFAM-DPCs

Purified TFAM-DPCs were diluted to 1 ng/μL on the basis of DNA concentration, followed by serial dilution using 20 mM HEPES (pH 7.4), 0.1 mM EDTA, SDS (0.1% *w/v*), and 1 ng/μL DNA (80 bp duplex DNA containing a dU). A standard solution of 30 μL and 120 μL of DNA coating solution was added to each well. Chemiluminescent intensities were plotted against the TFAM-DPC amounts (fmol). A range of TFAM-DPC amounts with the best $R^2$-value in linear regression was defined as the linear range of the detection. To evaluate the recovery of TFAM-DPCs, 4 fmol TFAM-DPCs were spiked into a mixture (50 μL) containing 100 ng λ DNA, 9 pmol recombinant TFAM in a solution containing 20 mM HEPES (pH 7.4), 0.1 mM EDTA, and SDS (0.1% *w/v*). The mixture was purified with the Zymo DNA purification kit, and the amount of recovered TFAM or TFAM-DPCs was obtained via chemiluminescence on a tissue culture plate (BrandTech, 81965). As a reference to the input amount, 4 fmol TFAM-DPCs were diluted to 120 μL using a solution containing 20 mM HEPES (pH 7.4), 0.1 mM EDTA, and SDS (0.1% *w/v*). For chemiluminescent detection, 30 μL of the sample and 120 μL of the commercial DNA-coating solution were added per well. The recovery was calculated by dividing the chemiluminescent intensity of the sample by that of the reference. The recovery of noncovalently associated

TFAM was assessed by mixing 9 pmol of free TFAM with 100 ng λ DNA, followed by purification and detection as described above.

### 2.6. Cell Cultures

HEK293 Tet-On cells expressing mitochondrial targeting a UNG-Y147A variant were prepared using a lentiviral-based approach as described previously [23]. Cells were cultured in T75-flasks containing DMEM supplemented with 4.5 g/L glucose, 10% (*v/v*) FBS, 100 mg/L G418, 2 mg/L puromycin, 1 mM pyruvate, and 0.1 g/L streptomycin/ampicillin. The culture was maintained at 37 °C under 5% $CO_2$. Once the confluence reached 60%, doxycycline was added to a final concentration of 2 µg/mL for the inducible overexpression of UNG1-Y147A. For untreated cells, an equal volume of sterilized water was added. Cells were lifted with a trypsin/EDTA solution and collected through centrifugation at $600\times g$ for 5 min. Cell pellets were resuspended with $1\times$DPBS (Dulbecco's phosphate buffered saline) and spun at $600\times g$ for 5 min. The washed cell pellets were flash frozen in liquid nitrogen and stored at $-80$ °C until further analysis.

### 2.7. Isolation of Mitochondria from Cultured Cells

HEK293 cells from 5 T-75 flasks (confluence 90%) were combined and resuspended in 5 mL of ice-cold cytosol extraction buffer containing 10 mM HEPES (pH 7.4), 320 mM sucrose, and 0.2% (*w/v*) BSA. The suspension was incubated on ice for 10 min, followed by manual homogenization with a 5 mL Dounce grinder on ice. One-hundred passes were required for HEK293 cells. The homogenate was centrifuged at $1200\times g$ for 10 min at 4 °C, followed by centrifugation of the supernature again at $1200\times g$ for 10 min at 4 °C. The supernatant was centrifuged at $15,000\times g$ for 20 min at 4 °C to pellet the crude mitochondria. Mitochondria were resuspended in 200 µL of DNase reaction buffer containing 20 mM HEPES (pH 7.4), 210 mM mannitol, 70 mM sucrose, 10 mM $MgCl_2$, 5 mM $CaCl_2$, and 0.2% BSA. The mitochondria suspension was incubated with 1000 U of turbonuclease at 37 °C for 20 min to remove nDNA contaminants. After nuclease digestion, mitochondria were washed twice with 1 mL of the cytosol extraction buffer and pelleted by centrifugation at $15,000\times g$ for 20 min at 4 °C. The purified mitochondria pellets were stored at $-80$ °C until further analysis.

### 2.8. Evaluation of mtDNA Purity

The amounts of mtDNA and nDNA were quantified using quantitative PCR with specific primers. The DNA concentration was determined by a pico-green-based assay. For nDNA, primers targeting the b-globin gene were used with 1 ng template DNA (forward primer: CGAGTAAGAGACCATTGTGGCAG, reverse primer: AGGGTCTTGGGTACAGGAGTT). For mtDNA, primers targeting the $tRNA_{glu}$ gene were used with 2 pg template per reaction (forward primer: CCCCACAAACCCCATTACTAAACCCA, reverse primer: TTTCATCAT-GCGGAGATGTTGGATGG). mtDNA template solutions were diluted with 0.1% (*w/v*) tween-20. The reaction was performed in a 10 µL solution containing 1X Phusion HF buffer, 200 µM dNTPs, 10 µM mixed primers, and 0.2 units of Phusion DNA polymerase on a Bio-Rad CFX Connect real-time PCR system. The reactions were held at 98 °C for 2 m, followed by a temperature program of 10 s at 98 °C, 30 s at 65 °C, and 30 s at 68 °C for 45 cycles. The Ct-values were calculated by the CFX Maestro Software (Bio-Rad). The melting temperature was analyzed after the reaction to ensure specific products were amplified. The absolute copy number was calculated according to the standard curve of each specific target. The purity of mtDNA was calculated with the following equation:

$$Purity(\%) = \frac{CN_{mt} \times 16500 \times 500}{CN_{mt} \times 16500 \times 500 + CN_n \times 3.3 \times 10^9} \times 100\% \qquad (1)$$

where $CN_{mt}$ is the absolute copy number of mitochondrial DNA. $CN_n$ is the absolute copy number of nuclear DNA. The equation was derived on the assumption that each set of human nuclear DNA is composed of 3.3 billion base pairs.

*2.9. Detection of TFAM-DPCs in mtDNA*

Isolated mitochondria were resuspended in 50 μL of ice-cold 1X PBS containing 60 μg of RNase A (NEB, T3018L) and 1X proteinase inhibitor cocktail. Mitochondria were lysed in a cell lysis buffer (50 μL) from the Monarch® genomic DNA purification kit (NEB, T3010). The mixture was incubated on ice for 10 min, followed by incubation with 0.1 M NaBH₄ for 2 min. The sample was mixed with 700 μL of DNA binding solution from the DNA Clean & Concentrator-5 kit (Zymo, D4014) and processed as described above. The concentration of DNA was measured using a pico-green-based assay. DNA was diluted to 0.5 ng/μL using a solution containing 20 mM HEPES (pH 7.4), 0.1 mM EDTA, and SDS (0.1% *w/v*). 60 μL of the diluted DNA and 90 μL of DNA coating solution were added to each well of a tissue culture microplate (BrandTech, 781965), followed by incubation at 4 °C for 22−24 h. The subsequent washing, antibody coating, and detection steps were the same as described in Section 2.4. Raw chemiluminescence was converted to the amount of TFAM-DPCs according to the standard curve. Chemiluminescent intensities were normalized to those of untreated samples to obtain the fold change.

## 3. Results

*3.1. Experimental Design*

We aimed to develop an ELISA-based method for detecting TFAM-DPCs in mtDNA from cultured human cells. To evaluate the purity and recovery yield specifically and quantitatively, we prepared TFAM-DPCs using recombinant human TFAM and model DNA substrates (Figure 1, step 1). We used model TFAM-DPCs mixed with a large excess of TFAM or DNA to optimize the purification (Figure 1, step 2) and detection procedures (Figure 1, step 3). The use of TFAM-DPC standards allowed us to differentiate between the signals of TFAM-DPCs and those of residue TFAM. For proof of principle, we detected TFAM-DPCs in mtDNA from HEK293 cells. The importance of mtDNA purity was evaluated, and a purification procedure for mtDNA with nearly no contaminating nDNA was developed (Figure 1, step 4).

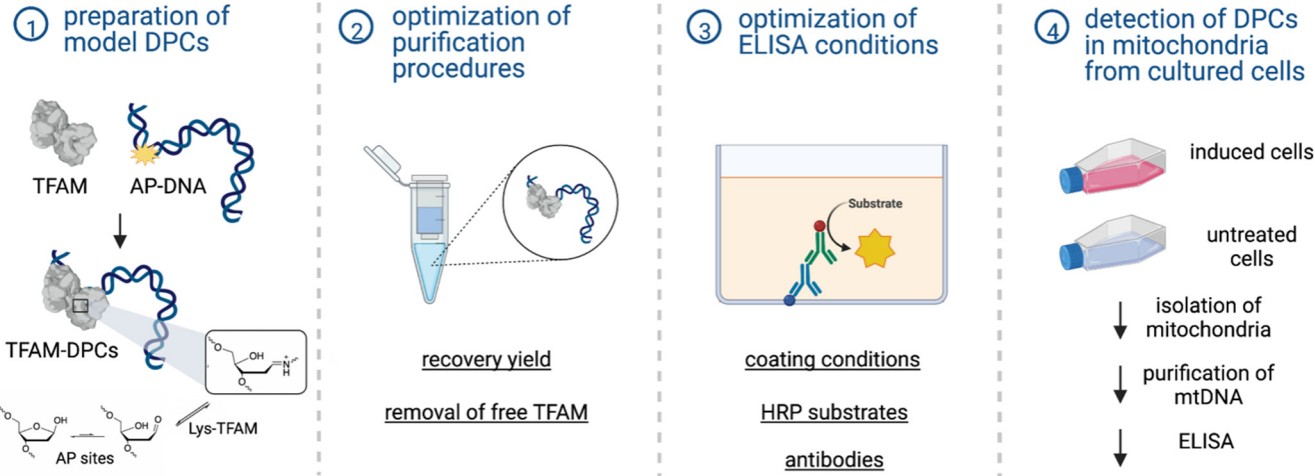

**Figure 1.** Overview of the experimental design. Purification and detection procedures were optimized using model TFAM-DPCs. The chemical reaction between Lys residues of TFAM and AP sites to form the Schiff base is shown in step 1. Key parameters for optimization are underlined in steps 2 and 3. For proof of concept, TFAM-DPCs were detected in HEK293 cells in step 4. The figure was created with BioRender.com (https://biorender.com/ access on 1 November 2022).

### 3.2. Preparation of Model TFAM-DPCs

To optimize the DPC isolation and detection workflow, we prepared model TFAM-DPCs from reactions with TFAM and AP-DNA. TFAM-DPCs were cross-linked via a reduced (by NaBH₃CN) Schiff base between TFAM and an 80 bp DNA substrate containing a site-specific AP lesion (hereinafter referred to as AP-DNA). AP-DNA was prepared from a deoxyuridine (dU)-containing oligomer treated with uracil DNA-glycosylase (UDG, Figure 2A) followed by annealing with a complementary oligomer. The presence of AP sites was confirmed by DNA cleavage under NaOH. Incubation of recombinant TFAM and AP-DNA at a molar ratio of 8 for 24 h resulted in quantitative conversion (>99% yield) of AP-DNA to TFAM-DPCs (Figures 2B and S1), as evidenced by the complete disappearance of AP-DNA. The smearing of product bands could be attributed to a partial denaturation of the DNA component of TFAM-DPCs due to the concomitant formation of single-strand breaks at AP sites (a mixture of DPCs with duplex DNA and a 6-mer oligomer flap). Such characteristics do not affect subsequent experiments. Due to the excess of TFAM in the reaction, the resulting product mixture contained TFAM-DPCs and free TFAM. The mixture was suitable for the evaluation of the extraction and purification yield.

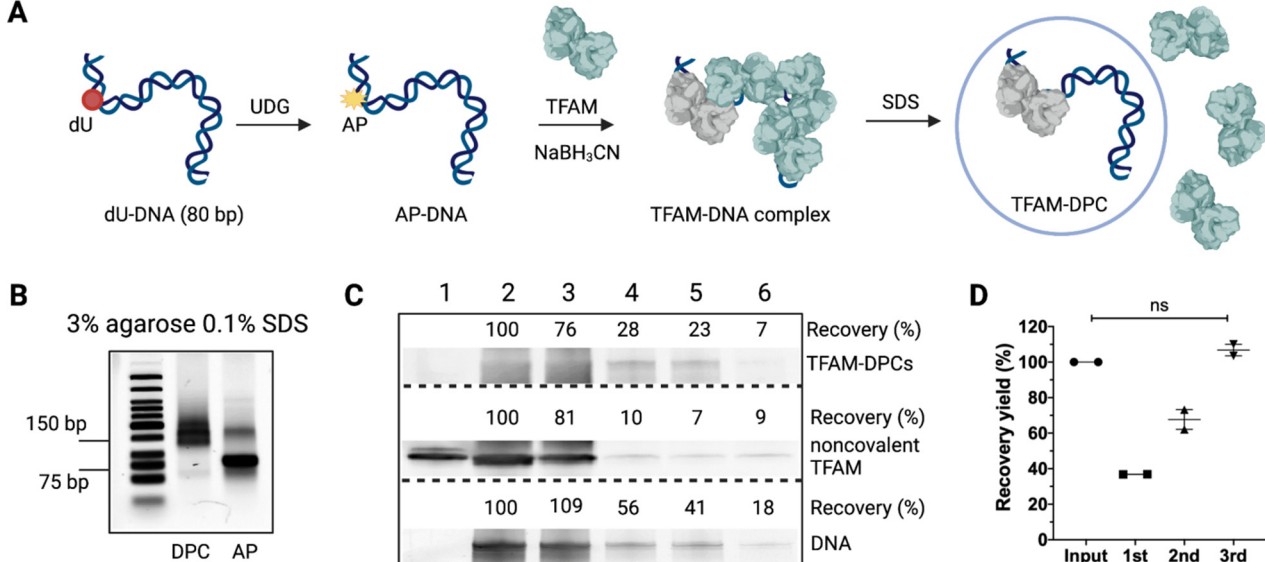

**Figure 2.** Preparation and purification of model TFAM-DPCs. (**A**) The workflow for preparing TFAM-DPCs. The procedures include the generation of AP sites through UDG digestion, cross-linking and reduction of Schiff base (by NaBH₃CN) in situ, and removal of noncovalently bound TFAM through washing with a sodium dodecyl sulfate (SDS)-containing solution. (**B**) SDS-agarose gel electrophoresis to verify the purity of the model TFAM-DPCs. The polymerized agarose gel was soaked in the running buffer containing 0.1% SDS followed by gel electrophoresis in the presence of the same running buffer. (**C**) Evaluation of the recovery yield of TFAM-DPCs and noncovalent TFAM associated with the DNA by western blotting. Recovery yields were derived from the product intensity relative to that of the input. Lane 1, recombinant TFAM; lane 2, input of TFAM and AP-DNA reaction mixture; lane 3, products purified by phenol/chloroform extraction; lane 4, products purified through ZYMO DNA purification and concentrator kit; lane 5, products purified through NEB PCR cleanup kit; lane 6, input processed through NEB genomic DNA purification kit. (**D**) Recovery yield of TFAM-DPCs in each load-wash-elute cycle using the ZYMO DNA purification and concentrator kit. The data were averaged from two independent experiments, and errors indicate the range of data.

### 3.3. Optimization of DPC Purification Procedures

Accurate detection of DPCs depends on highly efficient and specific removal of free, non-cross-linked proteins, such as TFAM. A major challenge lies in that the physiochemical properties of DPCs are different from pure DNA or proteins. Therefore, careful evaluation of different purification methods is necessary. Using a mixture of TFAM-DPCs and TFAM from TFAM-AP-DNA reactions, we assessed the recovery yield and the purity of TFAM-DPCs from several common purification procedures. We used western blotting to trace the noncovalent TFAM associated with the DNA and cross-linked TFAM, which can be separated during gel electrophoresis. We used fluorescence signals from SYBR™ Gold to estimate the remaining DNA in the sample. Because procedures involving potassium dodecyl sulfate precipitation had been shown to be unsuccessful in separating free proteins from DPCs [22], we excluded it from our current investigation. Phenol/chloroform extraction is a common method for the removal of proteins. Using this method, we found that the presence of free TFAM was apparent based on western blotting (Figure 2C, lane 3), although the DPC yield was high. The observation indicated that phenol/chloroform extraction is not effective in removing the free TFAM from the reaction mixture. On the other hand, purification with commercial DNA purification kits removed a majority of free TFAM, albeit with overall low recovery yields of DPCs (Figure 2C, lane 4, lane 5, and lane 6). This is not surprising because the purification kits are optimized for DNA, not for DPCs. However, we found that the DPC yield could be improved by loading the flow-through onto the spin column thrice to achieve a combined yield of >90% (Figure 2D). The ZYMO DNA purification and concentrator kit provided the highest recovery yield of TFAM-DPC among the three kits tested and was used in subsequent procedures. Therefore, efficient removal of free TFAM and a relatively high recovery yield for DPCs ensure the specificity of subsequent immunoassays.

### 3.4. Optimization of ELISA Conditions for the Detection of TFAM-DPCs

Compared with immunoblotting, ELISA typically has higher sensitivity and throughput, and requires a relatively low amount of materials [24–26]. Therefore, we chose ELISA as the detection platform. First, we evaluated the coating conditions with several types of plates and different coating solutions. With an ELISA high-binding plate and a phosphate buffer saline (PBS) coating solution, the absorbance ($A_{650nm}$) increased with the increasing amount of free TFAM or TFAM-DPCs (Figure 3A, left panel). No apparent difference in the signal intensity was observed between samples coated using PBS and those using a commercial DNA-coating solution. The signal intensities from TFAM and TFAM-DPCs were comparable, indicating that the ELISA plate could not distinguish cross-linked TFAM from noncovalent TFAM. This is problematic for TFAM, because its relatively high abundance and high DNA-binding affinity could result in noncovalent association with DNA, which could skew the results. Considering that the surface of the ELISA high-binding plate is designed for trapping proteins, we speculated that the treated surface might affect the performance of the DNA-coating solution. However, even with plates with an untreated polystyrene surface, the signals of TFAM-DPC-coated wells were indistinguishable from the background when the amount was less than 8 fmol per well (Figure S2). On the other hand, with a tissue culture plate, we observed a significant difference in signal intensities between TFAM-DPCs and TFAM (Figure 3A, right panel), although the overall absorbance was lower than that observed with an ELISA plate (Figure 3B, left panel). On a tissue culture plate, TFAM-DPC samples coated with DNA-coating solution, but not PBS, showed a difference in the absorbance signal relative to the noncovalent TFAM from 0.08 fmol to 8 fmol (Figure 3A, right panel). Notably, in terms of specificity, tissue culture plates used with the DNA coating solution showed the best performance in the tested concentration range (Figure 3B, right panel). Distinguishing TFAM-DPCs from residual noncovalently associated TFAM is particularly important, considering the abundance of TFAM among mitochondrial nucleoid proteins [27]. Therefore, tissue culture plates and a commercial DNA coating solution were used in subsequent procedures.

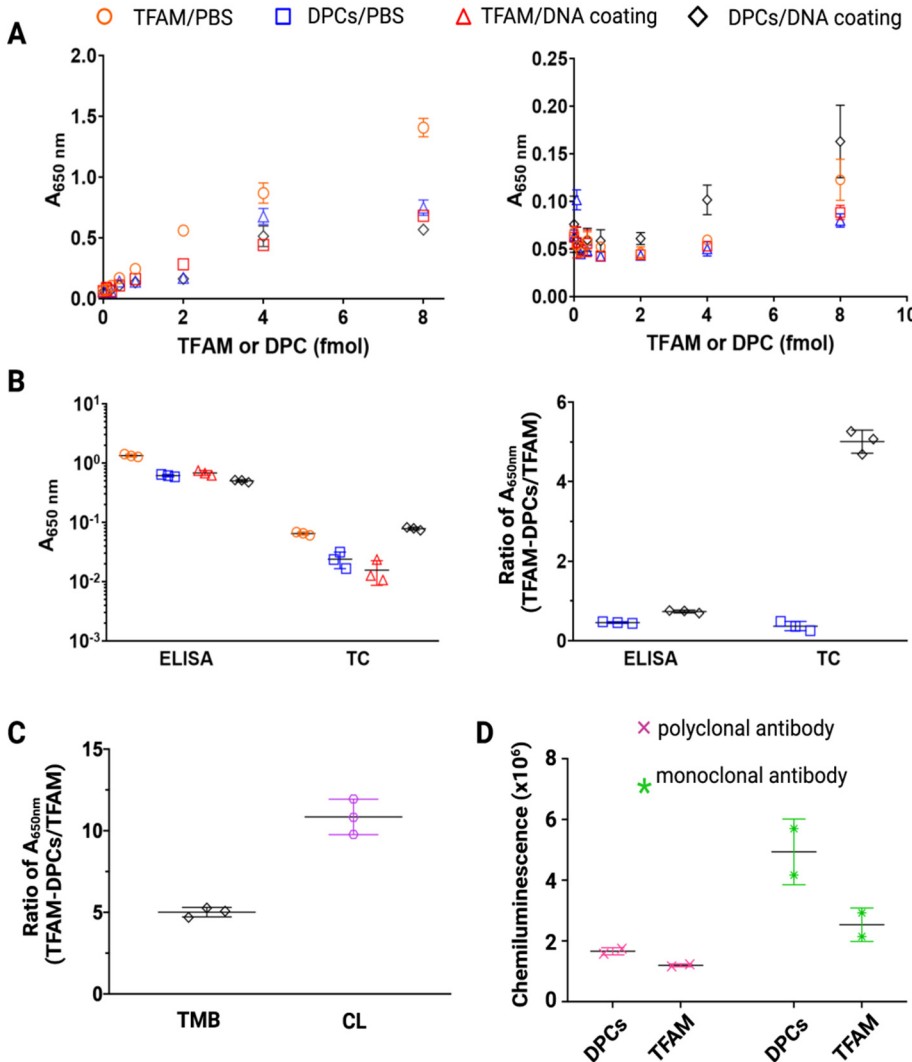

**Figure 3.** Optimization of ELISA conditions. (**A**) Colorimetric absorbance intensities at 650 nm measured using ELISA plates (left) or tissue culture (TC) plates (right). Orange circles indicate the absorbances with TFAM coated in a PBS solution; blue squares indicate the absorbances with TFAM-DPCs coated in a PBS solution; red triangles indicate the absorbances with TFAM coated in a commercial DNA-coating solution; black diamonds indicate the absorbances with TFAM-DPCs coated in commercial DNA-coating solution. (**B**) Comparison of the absorbances obtained from 8 fmol input TFAM or TFAM-DPCs under different coating conditions (left) and the fold-change of the absorbances normalized to TFAM input under the same coating condition (right). (**C**) Effects of horseradish peroxidase (HRP) substrates and primary antibodies. $A_{650nm}$ ratios of TFAM-DPCs/TFAM (8 fmol input) obtained from different HRP substrates (TMB or chemiluminescent). The absorbance or chemiluminescence (CL) was normalized to the amount of TFAM input under the same coating condition. (**D**) Chemiluminescence of TFAM and TFAM-DPCs with monoclonal or polyclonal primary anti-TFAM antibody. Data are mean, and errors represent deviation (n = 3).

Further, we evaluated the effects of HRP substrates and antibodies on sensitivity and specificity. Chemiluminescent substrates have been shown to provide better sensitivity than colorimetric substrates [28,29]. We compared a colorimetric HRP substrate (TMB) with a chemiluminescent substrate. Under 4 fmol or lower concentrations of TFAM or TFAM-DPCs, the signals of TFAM were indistinguishable from the background with PBS or the DNA coating solution on a tissue culture plate (Figure 3A, right panel). Under 8 fmol TFAM, the signal was significantly higher than the background. Therefore, we

discuss the specificity under 8 fmol TFAM or TFAM-DPCs input. Although the signal intensities were comparable between the two substrates with TFAM, a higher degree of specificity was observed with the chemiluminescent substrate (10 vs. 5 with TMB, Figure 3C). Using the chemiluminescent substrate, the detectable amount of TFAM-DPCs was 0.4 fmol per well. The lowest detectable amounts of TFAM-DPCs and free TFAM under different conditions were summarized in Table 1. Therefore, the chemiluminescent substrate achieved a higher specificity and lower detection limit compared with the colorimetric substrate.

**Table 1.** Summary of lowest detectable amounts of TFAM-DPCs and TFAM using different combinations of plates, coating solutions, and HRP substrates.

| | TFAM-DPCs (fmol) | free TFAM (fmol) |
|---|---|---|
| ELISA/PBS/TMB | 0.8 | 0.4 |
| ELISA/DNA coating/TMB | 2 | 0.4 |
| TC/PBS/TMB | >8 | 8 |
| TC/DNA coating/TMB | 8 | >8 |
| TC/DNA coating/chemiluminescent | 0.4 | 4 |

Lastly, we evaluated the signal intensities for TFAM-DPCs and noncovalent TFAM using primary antibodies from two different sources using western blotting. The TFAM monoclonal antibody was from Novus Biologicals, and the polyclonal primary antibody was from Cell Signaling Technologies. The polyclonal primary–secondary antibody pairs showed overall higher signal intensities and a stronger signal for DPCs relative to TFAM (Figure 3D). The overall higher signal intensities observed may be due to multiple epitope sites recognized, although additional factors may contribute to the observation. On the other hand, DPCs may affect the recognition at the site recognized by the monoclonal antibody. Therefore, the antibody pairs from cell signaling were used in subsequent experiments. Taken together, a combination of tissue culture plate, DNA coating solution, and the chemiluminescent substrate achieved a high specificity for TFAM-DPCs with a lowest detectable amount of 0.4 fmol of TFAM-DPCs.

### 3.5. Purification and Detection of TFAM-DPCs from Cultured Human Cells

We evaluated the linear range of the assay using purified TFAM-DPCs. TFAM-DPCs were purified using the ZYMO DNA purification and concentrator kit as described. The purity was higher than 90% according to western blotting. As shown in Figure 4B, the assay showed a linear range from 6.7 to 54 fmol per well with an $R^2$ of 0.94. To assess the recovery yield of TFAM-DPCs in the presence of excess DNA or TFAM, we prepared different mixtures using purified TFAM-DPCs mixed with recombinant TFAM, λ DNA and TFAM. The recovery yield was determined by the chemiluminescent intensity ratio of the mixture processed through the optimized purification procedures and the inputting amount of purified TFAM-DPCs without a sample workup. In a mixture of TFAM-DPCs, TFAM, and λ DNA, the DPC yield was 98 ± 11% (Figure 4C). In a mixture of TFAM and λ DNA, a negligible recovery (<3%) for noncovalent TFAM was observed, reaffirming the specificity of the purification and detection workflow. Notably, the mixture contained a TFAM/DNA molecular ratio of 3000 (9 pmol TFAM and ~ 3 fmol λ DNA), which is comparable to the estimated molar ratio of TFAM/DNA (1000–1700) in human cells [30]. Therefore, the 32-fold higher recovery yield of TFAM-DPCs than free TFAM (98% vs. 3%) together with 10-fold higher sensitivity (Table 1, 0.4 fmol for TFAM-DPCs vs. 4 fmol for TFAM) resulted in a discrimination factor of 320 for TFAM-DPCs. The specificity ensures minimal interference from residual TFAM in biological samples.

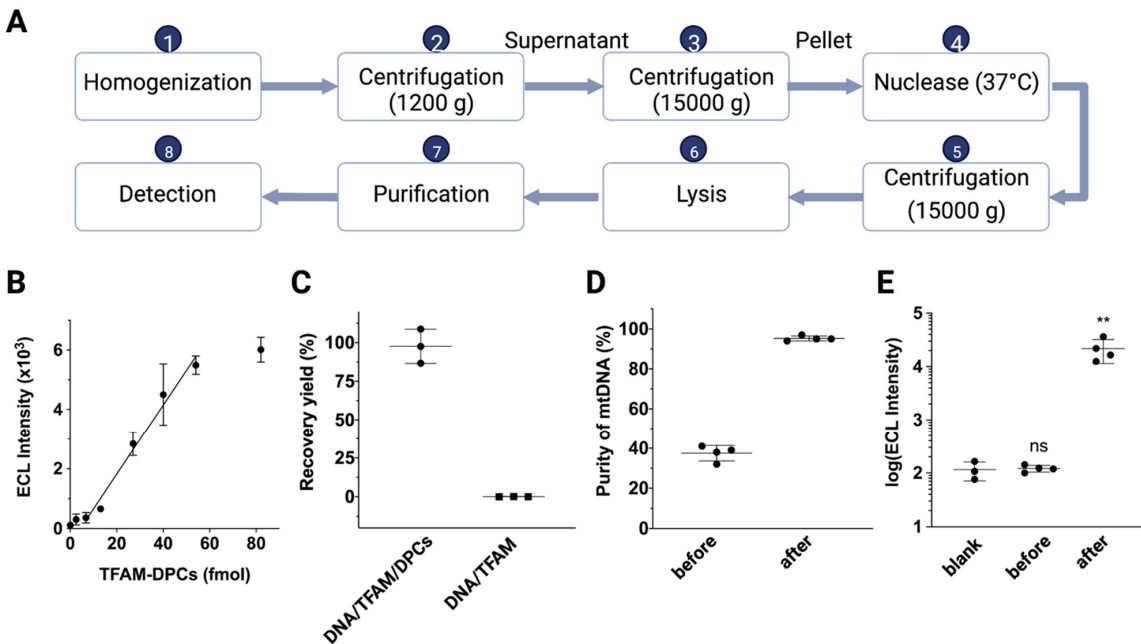

**Figure 4.** Purification and detection of TFAM-DPCs in mtDNA from cultured human cells. (**A**) Overview of the workflow for purifying and detecting TFAM-DPCs from cultured cells. (**B**) The linear range of chemiluminescent detection obtained with purified TFAM-DCPs. The amount of TFAM-DPCs in fmol was calculated based on the amount of AP-DNA in AP-DNA-TFAM reactions considering the quantitative conversion. (**C**) Recovery of TFAM-DPCs in a mixture of (i) TFAM-DPCs, TFAM and λ DNA and (ii) λ DNA and TFAM. (**D**) mtDNA purity from mitochondria isolated with a low-speed centrifugation method before and after the turbonuclease treatment. (**E**) Comparison of chemiluminescence obtained from mtDNA of HeLa cells with and without turbonuclease treatment. ns, no significance; ** $p < 0.001$.

A major challenge in working with mtDNA is the overall low abundance (~1%) of the mtDNA in a cellular DNA sample [6]. nDNA contamination can affect the reproducibility of the assay and interfere with data interpretation. Although density gradient-based methods have been used to obtain mtDNA, the methods require relatively large amounts of starting materials and a lengthy procedure and are limited in throughput [15]. On the other hand, low-speed centrifugation methods for isolating the mitochondria followed by lysis require a shorter procedure but typically provide a limited purity of mtDNA. Indeed, the purity of mtDNA from a low-speed centrifugation-based method was only 40% according to qPCR (Figure 4D), with the remaining being nDNA contaminants. To improve the purity, we included a turbonuclease digestion step during mitochondria isolation [31] and optimized the amount of nuclease to achieve the highest purity and yield of mtDNA. Digestion of contaminating nDNA before mitochondrial lysis ensures that mtDNA is protected from nuclease by the mitochondrial membranes under optimized conditions. Our results demonstrated that a 15 min treatment at 37 °C in the presence of turbonucleases was sufficient to achieve >95% purity for mtDNA (Figure 4D). While the amount of nDNA was reduced significantly, the amounts of mtDNA were maintained at similar levels according to qPCR-based quantification (Figure S3). Furthermore, when comparing the TFAM-DPC chemiluminescence signals obtained from mtDNA of HeLa cells with and without turbonuclease treatment, we noticed that the signal was significantly higher than the background in ELISA only when cells were treated with turbonuclease under optimized conditions (Figure 4E). These results indicated that the removal of contaminating nDNA from the mitochondrial DPC sample can improve the sensitivity of ELISA.

To validate the method for detecting TFAM-DPCs in cultured human cells, we prepared a cell line with an inducible level of AP sites in mtDNA. Tet-on HEK293 cells were

transduced with a lentiviral vector for inducible overexpression of mitochondria-targeting human UNG1 variant Y147A (UNG1-Y147A) [23]. The variant cleaves dT beside its native substrate dU [32]. Therefore, the expression of the variant can provide tunable levels of AP sites in mtDNA. Notably, AP sites are abundant DNA lesions that are present in mtDNA without UNG1-Y147A expression. Therefore, we intended to detect an elevated level of TFAM-DPCs under UNG1-Y147A induction. The overexpression of the UNG1 variant was confirmed by western blotting (Figure 5A). After treating cells with doxycycline for 24 h, a significant decrease in mtDNA copy number was observed (Figure 5B), as observed previously [23]. The results indicated that the damaged mtDNA containing AP sites were degraded rapidly. Our previous research demonstrated the role of TFAM in converting AP sites to strand breaks via TFAM-DPC intermediates [20]. To assess the accumulation of TFAM-DPCs, we purified TFAM-DPCs using the optimized workflow. Because of the fluctuation of the absolute signal readings each time, we normalized the signal based on a calibration curve each time with TFAM-DPC standards and obtained the amount of DPCs in fmol (Figure 5C). We measured the amount of TFAM-DPCs in the untreated and treated cells (Figure 5C) and observed a statistically significant increase in the level of TFAM-DPC under doxycycline treatment (Figure 5D). Therefore, our optimized purification and detection workflow is able to detect the change of TFAM-DPCs under different biological conditions.

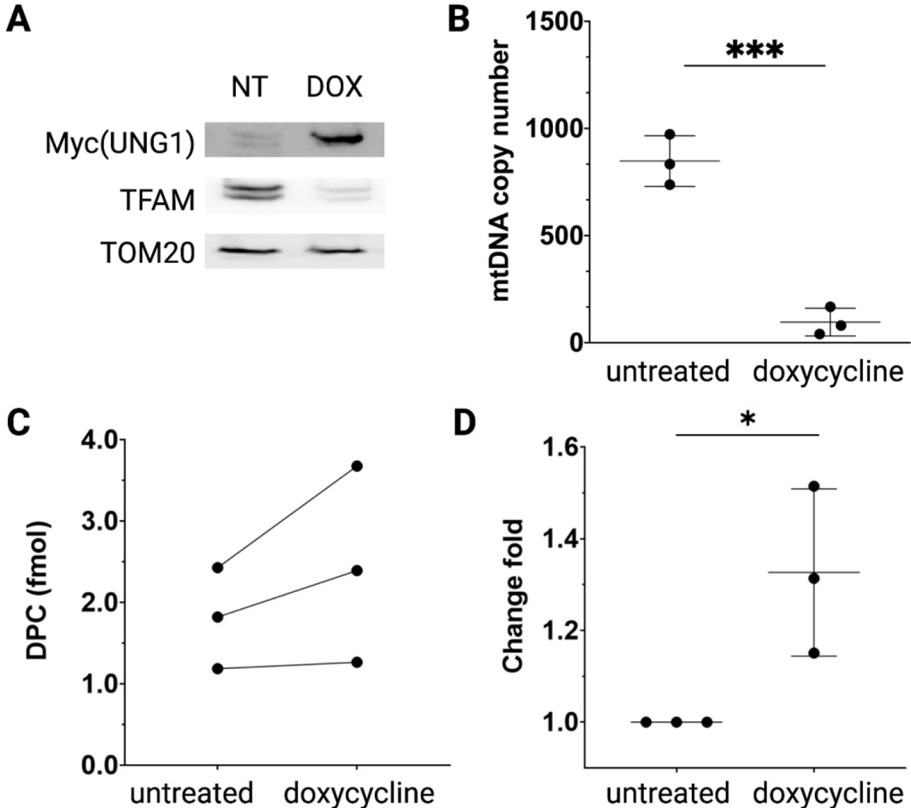

**Figure 5.** Detection of TFAM-DPCs in mtDNA from HEK293 cells with inducible mitochondrial targeting UNG1-Y147A. (**A**) Expression of Myc-tagged UNG1-Y147A confirmed by western blotting. (**B**) mtDNA copy number determined by normalizing to the copy number of nDNA using qPCR. (**C**) Quantification of TFAM-DPCs in the HEK293/UNG1-Y147A cells. (**D**) Fold change of TFAM-DPCs in doxycycline-treated HEK293/UNG1-Y147A cells relative to that from the untreated cells. * $p < 0.05$.

## 4. Discussion

We developed an ELISA-based method to measure the level of DPCs in mitochondria. Our new method is able to purify and quantify TFAM-DPCs with high specificity with minimal interference from the noncovalently associated TFAM. This is particularly important

for proteins in high abundance or with high DNA-binding affinity. The method is suitable for high-throughput applications and requires no prior knowledge of the exact DPC structures. The method is potentially applicable to detecting other DPCs with heterogeneous cross-links. Considering that the method relies on the recognition of TFAM in DPCs by an antibody, the difference in binding of the TFAM primary antibody to varying types of TFAM-DPCs could potentially introduce bias in the detection outcome. In addition, as with any applications with antibodies, their specificity needs to be carefully verified to ensure unbiased results. We envisage that the purification workflow can also be combined with mass spectrometry analysis when structural information is warranted.

The method is amenable to detecting other mitochondrial DPCs, such as topoisomerase DPCs and TDP1 DPCs. The method requires only 30 ng of DNA per well, which is about half of what is reported for RADAR assays [14–16]. In particular, our optimized ELISA based on chemiluminescence should be applicable to other types of DPCs. We discuss key parameters for those who wish to use the method for other mitochondrial DPCs. First, DPC standards are key to optimization. Although not every cross-linking reaction can achieve the high yield observed with TFAM and AP-DNA, the Schiff base chemistry is potentially applicable to other systems. For example, site-specific AP sites can be placed near a lysine residue based on the crystal structure to facilitate cross-linking. Additionally, DPCs are attainable via other types of cross-linking chemistry [1]. Second, because of the different chemical nature of the cross-linked protein and the size of the DNA component, the purification yield would need to be evaluated for the DNA purification kit of choice. Our loading of the TFAM-DPCs on the spin column thrice could provide a baseline for optimization. Third, the quality and specificity of the antibodies are worthy of consideration. Fourth, due to the difference in the number of mitochondria in different cell and tissue types, the amount of turbonuclease requires optimization. For example, we found that the HEK293 cells require a greater amount of turbonuclease to achieve a similar level of purity compared with HeLa cells (Figure S4). Last but not least, because TFAM-DPCs are present at different levels in different cell types, the amount of input DNA varies based on cell type. For example, 10 ng (per well) mtDNA coated onto the plate was sufficient for HEK293 cells, whereas 30 ng mtDNA was needed for HeLa cells (Xu et al. unpublished work).

In summary, our developed ELISA-based method showed excellent sensitivity and specificity for TFAM-DPCs in biological samples. With optimization, the method is expected to facilitate the detection and quantification of other types of mitochondrial DPCs.

**Supplementary Materials:** The following are available online at https://www.mdpi.com/article/10.3390/dna2040019/s1, Figure S1: cross-linking reactions between TFAM and 80bp AP-DNA monitored by agarose gel. Figure S2: the enhanced chemiluminescent signals obtained from a microplate with an untreated polystyrene surface; Figure S3: Ct-values from turbonuclease treated mitochondria samples; Figure S4: evaluation of mtDNA purity (based on qPCR data) under different amounts turbonuclease with HeLa cells and HEK293 cells.

**Author Contributions:** Conceptualization, L.Z. and W.X.; Data curation, W.X.; Formal analysis, W.X. and L.Z.; Funding acquisition, L.Z.; Methodology, W.X.; Project administration, L.Z.; Supervision, L.Z.; Validation, W.X. and L.Z.; Writing—original draft, W.X. and L.Z.; Writing—review & editing, L.Z. All authors have read and agreed to the published version of the manuscript.

**Funding:** This work was supported by National Institutes of Health (NIH) Grant R35 GM128854 (to L.Z.) and the University of California Riverside.

**Institutional Review Board Statement:** Not applicable.

**Informed Consent Statement:** Not applicable.

**Data Availability Statement:** All data were included in the manuscript and Supplementary Materials.

**Conflicts of Interest:** The authors declare no conflict of interest.

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
