# Peer review of "An Enzyme-Linked Immunosorbent Assay for the Detection of Mitochondrial DNA–Protein Cross-Links from Mammalian Cells"

_2673-8856, doi:10.3390/dna2040019_

Round 1

Reviewer 1 Report

Comments and Suggestions for Authors

Overall Comments:

The paper developed a new method to detect TFAM-DCPs in mtDNA, with potential wide application. Overall, it is well-written, with clear logic and scientific rigor, and thus meaningful to publish. Some suggested revisions are listed below. Briefly, the Introduction section needs to introduce the previously reported methods more explicitly with their pros and cons, which would help drive the need for developing a new method. Also, the potential limitations of this method should be clearly discussed in Discussion. In addition, some interpretation of the results should be clarified (see point-to-point comments below) and errors on select figure labels should be fixed.

Point-to-point Comments:

[Page 1 - line 35] “Other types of DPCs, such as topoisomerase DPCs, rely on designated enzymes to hydrolyze the covalent linkage between DNA and Protein” – This sentence is coming out of nowhere. The authors should more explicitly introduce different types of DPCs, and then their respective formation, repair, and biological significance.

[Page 2 - line 48-60] When introducing the previously reported methods, the authors should try to quantify the required amount of starting material and throughput.

[Page 2 - line 59] “Few methods are available for detecting mitochondrial DPCs” - The authors suggested that those previous methods for nuclear DPCs cannot be used for mtDCPs. Is that only because of the purification? Namely, the question is whether those previous “detection” methods for nDPC detection could also be used for mtDPCs if they had a suitable “purification” approach for mtDPCs? Think the answer is probably yes, as it has been briefly mentioned in the Discussion section. But the authors should also clearly cover this in the Introduction, so that the readers can clearly understand the existing development and the gap needed to bridge.

[Page 2 - line 64] “Detection of these DPCs has been achieved using a dot-blotting method based on immunodetection using either a specific antibody for the mitochondrial isoform of topoisomerase or after density gradient-based purification”—Here, need to introduce each method separately and more explicitly by providing pros and cons for each. And then drive to the reasons about why there are still needs to develop a new method that is presented in this paper.

[Page 2 - line 72] “The reaction proceeds via Schiff base chemistry, forming TFAM-DPCs.”—Would be good if the authors could provide a brief figure (chemical structures) for this reaction.

[Page 2 - line 74] “We chose an ELISA-based method to detect all forms of TFAM-DPCs collectively”—What is the evidence for the ability to detect ALL forms of TFAM-DPCs? Think the authors should attenuate the tone a bit when making this statement. For example, it is very likely that the primary antibody binding affinity can be different across different forms of TFAM-DPCs or different mtDNA sequences surrounding the DPC sites, which could introduce bias on the detection result. And this should be discussed clearly as a potential limitation in the Discussion section.

[Page 2 - line 75]We presume that a mass spectrometry-based approach can potentially lead to biased results due to different recovery yields and ionization efficiencies of the peptides, albeit powerful in detecting other homogenous DPC types [22].” – Need to provide literature reference to support the bias of mass-spectrometry-based methods. The current reference [20] is actually to support that the mass-spectrometry-based methods can provide “unbiased” assessment of the whole protein DPC adductome. In addition, when making this statement, were the authors trying to say the method developed in this paper has less bias than the mass-spectrometry-based approaches? If so, what is the evidence? If not, need to make it clear so that the readers would not interpret it that way.

[Page 7 - line 294]Incubation of recombinant TFAM and AP-DNA at a molar ratio of 8 for 24 h resulted in quantitative conversion (>99% yield) of AP-DNA to TFAM-DPCs (Fig. 2B)” – The authors should try to explain the multiple bands on the gel. Also, how could the authors know that all AP-DNA form covalent bonds with TFAM (vs. noncovalent association)?

[Page 7 - line 306] “We used western blotting to trace the noncovalently associated TFAM and cross-linked TFAM, which can be separated during gel electrophoresis.” - So, in Figure 2C, do the bands labeled as “Free FTAM” include the FTAM that are noncovalently associated with the DNA. If so (which would make sense), the authors should make it clear in the text.

[Page 8 - line 318] “However, we found that the DPC yield could be improved by loading the flow-through onto the spin column thrice to achieve a combined yield of > 90% (Fig. 2D).”—How about the recovery of free TFAM when this was being done? It should be noted in the text.

[Page 9 - line 377] “The chemiluminescent substrate achieved a higher specificity and lower detection limit compared to the colorimetric substrate.” – So, the recovery rates for TFAM-DPCs are 3 times higher than free TFAM. [Page 10 - line 393] “Taken together, a combination of tissue culture plate, DNA coating solution, and the chemiluminescent substrate achieved a high specificity for TFAM-DPCs with a lowest detectable amount of 0.4 fmol of TFAM-DPCs.” – So, the sensitivity to FTAM-DPCs is ~10 times higher than free FTAM in the optimized ELISA condition. Therefore, taken together, the overall specificity of “FTAM-DPCs” over “free FTAM” is ~30%. Would that be enough to accurately detect FTAM-DPCs from cells (given the amount of free FTAM vs. FRAM-DPCs in cells)? The authors should comment on this in the main text.

[Page 10 - line 389] “(Fig. 3C, right panel)”—There is no Fig.3C right panel. It should be Fig. 3D in the figure. Should fix this.

[Page 11 - line 405] “we evaluated the linear range of the assay using purified TFAM-DPCs.” – for samples in Fig.4B, the authors should note how they determined the exact fmol of the purified TFAM-DPCs.

[Page 11 - line 420] “In a biological matrix with TFAM-DPCs and mitochondrial lysates, the recovery was 190 ± 20 % (Fig. 4C). Given the high specificity observed with mixtures, we presume that the signal was contributed by endogenous TFAM-DPCs (vide infra).” – Then, what is the purpose of running this sample? What is the hypothesis? What is the implication of this result?

[Page 11 - line 435] “Our results demonstrate that a 15-min treatment at 37 °C in the presence of Turbo nucleases was sufficient to achieve >95% purity for mtDNA.”  - How does Turbonuclease treatment impact the yield of mtDNA? Fig.5E shows the result. But the authors should also comment on it in the main text. Should also explain why Turbonuclease seems to have larger tendency to degrade nDNA vs. mtDNA

[Fig 5B] Is that the result from “UNG1-Y147A” cell? May also show the result of the control (wt cells)

[Page 12 - Line 459] “We compared the amount of TFAM-DPCs in the untreated and treated pairs (Fig. 5C) and detected a statistically significant increase in cells with induced AP sites (Fig. 5D)”—Did the authors want to make a statement that TFAM-DPCs levels increase after doxycycline treatment? The authors should comment on how conclusive the result is, and proactively provide what could be alternative reasons for the enhanced observed signal in the treated sample.

[Page 13 - line 471] “The method is applicable to detecting DPCs with heterogeneous cross-links.” The authors should comment on potential binding affinity difference between the primary antibody and different FTAM-DCPs. (e.g., DNA sequence dependence). Beyond that, in general, the limitations of this method should be clearly and objectively discussed in the Discussion section. (e.g., When using column / Turbo nuclease to remove nDNA, there could be bias between degrading mtDNA and mtDNA-PDC)

[Figure 3A] The key (shape and color) is not consistent with the labels on the plot

[Figure 3B (left panel)] The values are not consistent with those in 3A.

[Figure 3B (right panel)] The y-axis (“DPCs/TFAM”) is wrong and mis-leading. Also, the plot may not show the 4 bars of “TFAM/TFAM (whose values are of course = 1)”. Showing them on the plot just creates confusion to readers. (Same comment for 3C)

[Table 1] Should include the way of calculating the detection limit in the Methods section

Author Response

We thank the reviewer for his/her careful attention to our manuscript and many helpful suggestions. We have revised the manuscript accordingly with the point-by-point response listed below.

1. [Page 1 - line 35] “Other types of DPCs, such as topoisomerase DPCs, rely on designated

enzymes to hydrolyze the covalent linkage between DNA and Protein” – This sentence is coming out of nowhere. The authors should more explicitly introduce different types of DPCs, and then their respective formation, repair, and biological significance.

We have rewritten the sentence to improve the flow. It reads “Certain types of DPCs can also be resolved by the enzymatic hydrolysis of the covalent linkage between DNA and protein [3]. A prominent example is topoisomerase-mediated DPCs with their cognate repair enzymes, tyrosyl-DNA phosphodiesterases [2, 3].”

Also, in the same paragraph, we have summarized different types of DPCs and referred interested readers to several recent reviews, which contain detailed discussions on the formation and repair of DPCs. The added sentence reads “Consequently, DPCs represent a diverse class of DNA lesions, which differ in the protein component, the flanking DNA structure, and the cross-linking chemistry. Several recent reviews have summarized the different types of DPCs and their repair pathways in detail [2, 3].”

2. [Page 2 - line 48-60] When introducing the previously reported methods, the authors should try to quantify the required amount of starting material and throughput.

We thank the reviewer for the suggestion and have added the required number of cells for the methods.

3. [Page 2 - line 59] “Few methods are available for detecting mitochondrial DPCs” - The authors suggested that those previous methods for nuclear DPCs cannot be used for mtDCPs. Is that only because of the purification? Namely, the question is whether those previous “detection” methods for nDPC detection could also be used for mtDPCs if they had a suitable “purification” approach for mtDPCs? Think the answer is probably yes, as it has been briefly mentioned in the Discussion section. But the authors should also clearly cover this in the Introduction, so that the readers can clearly understand the existing development and the gap needed to bridge.

We thank the reviewer for comments. Previous detection methods are applicable to detecting mitochondrial DPCs theoretically. However, considering the overall low abundance of mtDNA, the detection step also requires optimization and validation. We have clarified this point on Page 2.

4. [Page 2 - line 64] “Detection of these DPCs has been achieved using a dot-blotting method

based on immunodetection using either a specific antibody for the mitochondrial isoform of

topoisomerase or after density gradient-based purification”—Here, need to introduce each

method separately and more explicitly by providing pros and cons for each. And then drive to the reasons about why there are still needs to develop a new method that is presented in this paper.

We thank the reviewer for the suggestion and have discussed the limitation of the method at the end of the 2nd paragraph on Page 2.

5. [Page 2 - line 72] “The reaction proceeds via Schiff base chemistry, forming TFAM-DPCs.”

—Would be good if the authors could provide a brief figure (chemical structures) for this reaction.

We thank the reviewer for the suggestion. We have added the chemical structures in Figure 1.

6. [Page 2 - line 74] “We chose an ELISA-based method to detect all forms of TFAM-DPCs

collectively”—What is the evidence for the ability to detect ALL forms of TFAM-DPCs? Think the

authors should attenuate the tone a bit when making this statement. For example, it is very likely that the primary antibody binding affinity can be different across different forms of TFAM-DPCs or different mtDNA sequences surrounding the DPC sites, which could introduce bias on the detection result. And this should be discussed clearly as a potential limitation in the Discussion section.

We thank the reviewer for the suggestion and have revised the relevant sections in Introduction and Discussion. In Introduction, the sentence now reads “We chose an ELISA-based method due to its high throughput and the relatively low amount of samples required. The latter reason is particularly important for mtDNA associated DPCs because of the low abundance of mtDNA in cellular samples and the low stoichiometric ratios of cross-linked proteins relative to mtDNA.”

In Disucssion, we discussed the limitation of the method. It reads “The method is potentially applicable to detecting DPCs with heterogeneous cross-links. Considering that the method relies on recognition of TFAM in DPCs by an antibody, the difference in binding of the TFAM primary antibody to varying types of TFAM-DPCs could potentially introduce bias in the detection outcome. In addition, as with any applications with antibodies, their specificity needs to be carefully verified to ensure unbiased results.”

7. [Page 2 - line 75] “We presume that a mass spectrometry-based approach can potentially lead to biased results due to different recovery yields and ionization efficiencies of the peptides, albeit powerful in detecting other homogenous DPC types [22].” – Need to provide literature reference to support the bias of mass-spectrometry-based methods. The current reference [20] is actually to support that the mass-spectrometry-based methods can provide “unbiased” assessment of the whole protein DPC adductome. In addition, when making this statement, were the authors trying to say the method developed in this paper has less bias than the mass-spectrometry-based approaches? If so, what is the evidence? If not, need to make it clear so that the readers would not interpret it that way.

Our main reason for choosing an ELISA-based method is its high throughput and the relatively low amount of samples required. These reasons have been included in the revised text, as described in our answer to the previous question.

We appreciate the usefulness of mass spectrometry-based methods in detecting different DPCs. However, due to the heterogeneity of the TFAM-DPCs, we think it could be challenging to use mass spectrometry-based methods to quantify all forms of TFAM-DPCs due to the different recovery yields and ionization efficiencies of the resulting peptides. We have added the explanations to the revised text.

“TFAM-induced DPCs (and perhaps other proteins) are a heterogenous mixture of DPCs mediated by different Lys residues [21]. For this reason, although mass spectrometry-based approaches are powerful in detecting specific types DPCs [22], the heterogeneity of TFAM-DPCs would make mass spectrometry-based quantitation challenging due to different recovery yields and ionization efficiencies of the peptides.”

8. [Page 7 - line 294] “Incubation of recombinant TFAM and AP-DNA at a molar ratio of 8 for 24 h resulted in quantitative conversion (>99% yield) of AP-DNA to TFAM-DPCs (Fig. 2B)” – The authors should try to explain the multiple bands on the gel. Also, how could the authors know that all AP-DNA form covalent bonds with TFAM (vs. noncovalent association)?

We have added explanations for the smearing of product bands on Page 8. It reads “The smearing of product bands could be attributed to a partial denaturation of the DNA component of TFAM-DPCs due to the concomitant formation of single-strand breaks at AP sites (a mixture of DPCs with duplex DNA and a 6-mer oligomer flap). Such characteristics do not affect subsequent experiments.”

Regarding the second question, in our optimization of the gel conditions, we have carried out reaction time courses (0 to 36 h) and compared carefully different concentrations of SDS in stripping off noncovalently associated TFAM. As shown in Figure S1, without SDS, we observed several discrete bands throughout the reaction time course, indicating the presence of noncovalent TFAM-DNA complexes. However, in the presence of 1%, we observed the disappearance of the DNA substrates and production of a single product band, indicative of the completion of the reaction. We found that 0.1% is sufficient to disrupt all noncovalently associated TFAM and used 0.1% in subsequent experiments.

9. [Page 7 - line 306] “We used western blotting to trace the noncovalently associated TFAM and cross-linked TFAM, which can be separated during gel electrophoresis.” - So, in Figure 2C, do the bands labeled as “Free FTAM” include the FTAM that are noncovalently associated with the DNA. If so (which would make sense), the authors should make it clear in the text.

We thank the reviewer for the suggestion and have revised the text on Page 9. It reads “We used western blotting to trace the noncovalent TFAM associated with the DNA and cross-linked TFAM,…”

10. [Page 8 - line 318] “However, we found that the DPC yield could be improved by loading the flowthrough onto the spin column thrice to achieve a combined yield of > 90% (Fig. 2D).”—How about the recovery of free TFAM when this was being done? It should be noted in the text.

We have obtained the yield of both TFAM-DPCs and noncovalent TFAM using a mixture of TFAM-DPCs, TFAM, and λ DNA with a TFAM/DNA molecular ratio of 3000 (9 pmol TFAM and ~ 3 fmol λ DNA). After purifying the sample using the optimized workflow (loading the sample thrice on the column), the yields of TFAM-DPCs and noncovalent TFAM were 98% and 3%, respectively, as shown in Fig. 4C.

11. [Page 9 - line 377] “The chemiluminescent substrate achieved a higher specificity and lower detection limit compared to the colorimetric substrate.” – So, the recovery rates for TFAM-DPCs are 3 times higher than free TFAM. [Page 10 - line 393] “Taken together, a combination of tissue culture plate, DNA coating solution, and the chemiluminescent substrate achieved a high specificity for TFAM-DPCs with a lowest detectable amount of 0.4 fmol of TFAM-DPCs.” – So, the sensitivity to FTAM-DPCs is ~10 times higher than free FTAM in the optimized ELISA condition. Therefore, taken together, the overall specificity of “FTAM-DPCs” over “free FTAM” is ~30%. Would that be enough to accurately detect FTAM-DPCs from cells (given the amount of free FTAM vs. FRAM-DPCs in cells)? The authors should comment on this in the main text.

We appreciate the reviewer’s concern; however, there might be some misunderstanding of our data. As the reviewer pointed out that the sensitivity of TFAM-DPCs is 10 times higher than free TFAM in the optimized ELISA condition (Table 1). The recovery yield of TFAM-DPCs is 32-fold higher than free TFAM (98% vs 3%). Therefore, the discrimination of TFAM-DPCs is 320-fold higher compared to noncovalent TFAM. We have added a sentence on Page 11 to make it clear to the readers.

12. [Page 10 - line 389] “(Fig. 3C, right panel)”—There is no Fig.3C right panel. It should be Fig. 3D in the figure. Should fix this.

We thank the reviewer for the comments and have mixed the in-text citations accordingly.

13. [Page 11 - line 405] “we evaluated the linear range of the assay using purified TFAM-DPCs.” – for samples in Fig.4B, the authors should note how they determined the exact fmol of the purified TFAM-DPCs.

The amount of TFAM-DPCs in fmol was calculated based on the amount of AP-DNA in AP-DNA-TFAM reactions considering the quantitative conversion. We have included this statement in the Figure 4B caption.

14. [Page 11 - line 420] “In a biological matrix with TFAM-DPCs and mitochondrial lysates, the recovery was 190 ± 20 % (Fig. 4C). Given the high specificity observed with mixtures, we presume that the signal was contributed by endogenous TFAM-DPCs (vide infra).” – Then, what is the purpose of running this sample? What is the hypothesis? What is the implication of this result?

The purpose of this experiment was to see whether the performance of purification and detection workflow can be affected biological matrix. We agree with the reviewer that the results can be attributed to different reasons and therefore decided to omit the data from the manuscript. Figure 4C has been revised accordingly.

15. [Page 11 - line 435] “Our results demonstrate that a 15-min treatment at 37 °C in the presence of Turbo nucleases was sufficient to achieve >95% purity for mtDNA.” - How does turbonuclease treatment impact the yield of mtDNA? Fig.5E shows the result. But the authors should also comment on it in the main text. Should also explain why Turbonuclease seems to have larger tendency to degrade nDNA vs. mtDNA

We have included additional qPCR data (Figure S2) to demonstrate that while the amount of nDNA was reduced significantly, the amounts of mtDNA were maintained at similar levels according to qPCR-based quantification, achieving > 95% purity for mtDNA.

16. [Fig 5B] Is that the result from “UNG1-Y147A” cell? May also show the result of the control (wt cells).

Yes. We added HEK293/UNG1-Y147A cells in the Fig 5D captions. As for the wide-type UNG1 transduced cells, previous research has verified that these cells do not generate detectable mtDNA damage (Ref. 23). Therefore, we chose to use uninduced cells as controls, not the wide-type UNG1 transduced cells.

17. [Page 12 - Line 459] “We compared the amount of TFAM-DPCs in the untreated and treated

pairs (Fig. 5C) and detected a statistically significant increase in cells with induced AP sites (Fig.

5D)”—Did the authors want to make a statement that TFAM-DPCs levels increase after

doxycycline treatment? The authors should comment on how conclusive the result is, and

proactively provide what could be alternative reasons for the enhanced observed signal in the

treated sample.

Yes. We intended to state that TFAM-DPC levels increased after doxycycline treatment. We have another manuscript in revision with Nucleic Acid Research, which we have included in the original submission for review only. In the other manuscript, we have verified that the doxycycline treatment induced the AP site level, and also observed similar increase of TFAM-DPC levels in HeLa cells under doxycycline treatment. We discussed the biological context of the TFAM-DPCs in detail. In the present manuscript, we included the data for HEK293 cells to demonstrate the usefulness of the method.

18. [Page 13 - line 471] “The method is applicable to detecting DPCs with heterogeneous crosslinks.” The authors should comment on potential binding affinity difference between the primary antibody and different FTAM-DCPs. (e.g., DNA sequence dependence). Beyond that, in general, the limitations of this method should be clearly and objectively discussed in the Discussion section. (e.g., When using column / Turbo nuclease to remove nDNA, there could be bias between degrading mtDNA and mtDNA-PDC)

We thank the reviewer for the suggestion and have added discussion on page 15. It reads “The method is potentially applicable to detecting other DPCs with heterogeneous cross-links. Considering that the method relies on recognition of TFAM in DPCs by an antibody, difference in binding of the TFAM primary antibody to varying types of TFAM-DPCs could potentially introduce bias in the detection outcome. In addition, as with any applications with antibodies, their specificity needs to be carefully verified to ensure unbiased results.”

19. [Figure 3A] The key (shape and color) is not consistent with the labels on the plot [Figure 3B (left panel)] The values are not consistent with those in 3A.

We thank the reviewer for the suggestion and have revised Figure 3 accordingly.

20. [Figure 3B (right panel)] The y-axis (“DPCs/TFAM”) is wrong and mis-leading. Also, the plot may not show the 4 bars of “TFAM/TFAM (whose values are of course = 1)”. Showing them on the plot just creates confusion to readers. (Same comment for 3C)

We thank the reviewer for the suggestion and have revised Figure 3 accordingly.

21. [Table 1] Should include the way of calculating the detection limit in the Methods section.

We thank the reviewer for the suggestion and have added the following statement on Page 4. “The lowest detectable amounts of TFAM-DPCs and TFAM were defined as fmol of TFAM (or TFAM-DPCs) when the signal intensity was three times of that of the background.”

Reviewer 2 Report

Comments and Suggestions for Authors

Xu and Zhao report on the development of an ELISA protocol for the detection of mitochondrial DNA-protein cross-links. They rigorously optimized each step of the protocol before applying it to mtDNA from HEK cells. The work is solid and well done, the results are great, and the conclusions are supported by the data collected. I am fully in favor of publication upon the authors considering the few suggestions I have for them below.

It seems strange to have yields greater than 100% as reported in Figure 4C and the associated text in the first paragraph on page 11. Why are the yields greater than 100%?

Could the 5` side of the DNA sequences used be labeled for clarity (lines 110-114 and 227-230).

The compound sodium cyanoborohydride according to IUPAC nomenclature has the molecular formula NaBH3CN, not NaCNBH3 (lines 129 and 290 and Fig 2). This is a strange one in that the way we say it is not how we write it.

Author Response

We thank the reviewer for his/her careful attention to our manuscript and many helpful suggestions. We have revised the manuscript accordingly with the point-by-point response listed below.

1. It seems strange to have yields greater than 100% as reported in Figure 4C and the associatedtext in the first paragraph on page 11. Why are the yields greater than 100%?

Although the purpose of this experiment was to see whether the performance of purification and detection workflow can be affected biological matrix, we agree with the reviewer that the results can be attributed to different reasons and therefore decided to omit the data from the manuscript. Figure 4C has been revised accordingly.

2. Could the 5` side of the DNA sequences used be labeled for clarity (lines 110-114 and 227-230).

We thank the reviewer for the suggestion and have added 5’ to the DNA sequences accordingly.

3. The compound sodium cyanoborohydride according to IUPAC nomenclature has the molecular formula NaBH3CN, not NaCNBH3 (lines 129 and 290 and Fig 2). This is a strange one in that the way we say it is not how we write it.

We thank the reviewer for the suggestion and have corrected inconsistent terminologies accordingly.
